

# Global ionospheric sporadic E intensity prediction from GNSS RO using a novel stacking machine learning method incorporated with physical observations

Tianyang Hu[1], Xiaohua Xu[1,2], Jia Luo[1,3], Jialiang Hou[1], Haifeng Liu[4]

[1]School of Geodesy and Geomatics, Wuhan University, Wuhan, 430079, China
[2]Collaborative Innovation Center for Geospatial Technology, Wuhan, 430079, China
[3]Key Laboratory of Geospace Environment and Geodesy, Ministry of Education, Wuhan, 430079, China
[4]School of Surveying and Geoinformation Engineering, East China University of Technology, Nanchang, 330013, China

*Correspondence to*: Xiaohua Xu (xhxu@sgg.whu.edu.cn)

**Abstract.** Sporadic E (Es) layers, the irregularities of enhanced electron density that commonly occur in the ionospheric E region, are affected by the interactions between distinct atmospheric layers. Es intensity (EsI) is a crucial parameter to describe Es layer characteristics, while there still lacks the method for high-precision EsI prediction due to its complex spatiotemporal variation and physical driving mechanisms. We propose a novel stacking machine learning (SML) method for global EsI prediction, in which the EsI predicted by each base model are optimally integrated by the meta model to obtain reduced bias and variance. Various Es-related physical observations are incorporated as the inputs of SML together with the EsI derived from global navigation satellite system (GNSS) radio occultation (RO) measurements. SML performs well in both long-term and short-term EsI predictions and characteristics reconstruction. The SML-predicted EsI is in good agreement with the GNSS RO-derived EsI, with the mean error (ME) of 0.032 TECU km$^{-1}$ and root mean square error (RMSE) of 0.158 TECU km$^{-1}$. Taking ionosonde observations as reference, SML has the RMSE of 1.064 MHz, which is reduced by 20.1%–40.5% compared to existing prediction methods. The higher accuracy of our method than those not incorporating physical observations illustrates the significance of considering multiple related physical factors when constructing the Es prediction model. The proposed method can be expected to provide valuable information for not only ionospheric irregularities monitoring and space weather forecasting, but also the mechanisms of Es layer formation and atmospheric coupling.

# 1 Introduction

Ionospheric sporadic E (Es) layers are thin-layer structures with abnormally sharp enhanced densities of electrons and metal ions, occurring frequently in ionospheric E region with a major altitude range of 90–130 km. Existing studies show that the occurrence of Es layers is driven by various physical mechanisms in the lower atmosphere, mesosphere-lower thermosphere (MLT), ionosphere, and space environment, such as the neutral wind shear (Chu et al., 2014), upward propagating gravity waves (GWs) (Qiu et al., 2023), atmospheric tides (Tang et al., 2022a), solar activity (Yu et al., 2021), and geomagnetic field





(Luo et al., 2021a), resulting in its highly uncertain and irregular spatial and temporal characteristics. Specifically, the neutral wind shear theory is widely accepted for the Es layer formation in mid-latitudes, the low latitude Es layers are well related to wind shear and equatorial electrojet (Raghavarao et al., 2002), while the geomagnetic activity and vertical motion of gravity waves are more efficient in concentrating the ions of Es layers (Kirkwood and Nilsson, 2000; MacDougall et al., 2000a, b).

Therefore, the high-precision modeling and prediction of Es layers is not only crucial for ionospheric irregularities monitoring and space weather forecasting, but also provides solutions for understanding the mechanisms of Es layer formation and coupling of distinct atmospheric layers, while it still remains a challenging task due to its complex patterns and influencing factors.

Traditionally, Es layers are detected by ground-based devices such as ionosonde and incoherent scatter radar

(Leighton et al., 1962; Mathews, 1998; Heinselman et al., 2000). With the development of global navigation satellite system (GNSS) over the past two decades, the space-borne GNSS radio occultation (RO) missions have been widely applied to investigate the climatology of Es occurrence rates and Es intensity (EsI) in recent years (Arras et al., 2008; Chu et al., 2014; Yu et al., 2019; Xu et al., 2022; Liu et al., 2024a). Most RO missions consist of one or several low earth orbit (LEO) satellites, which can acquire all-weather and wide-area ionospheric observations with high vertical resolution. Based on the

large amount of accumulated RO data, some empirical models were established using statistical method to describe the global EsI distribution (Hu et al., 2022; Yu et al., 2022; Niu and Fang, 2023). However, since the high uncertainty and irregularity of EsI, these empirical models are difficult to predict the localized and short-term Es variations. Besides, these numerically established models are not incorporated with physical observations, thus cannot estimate EsI based on the Es-related physical mechanisms. To overcome this limitation, some scholars compared the morphologies of RO-derived EsI and

physical data to analyze their relationships (Qiu et al., 2019; Yu et al., 2019; Yamazaki et al., 2022), but these analyses were only qualitative and did not quantitatively reveal their correlations.

Recently, the advances in artificial intelligence algorithms provide new perspectives for data analysis in geosciences and many other fields. Machine learning (ML) is a powerful tool fitting complex nonlinear relationships between multiple variables, which has been employed to resolve the regression and classification problems in geoscience and is proven to have

better performance compared with traditional methods. Currently, there are few studies on EsI prediction based on ML methods. Although Emmons et al. (2023) and Tian et al. (2023) used ML models for the detection and reconstruction of Es layers, there were little data on Es-related physical mechanisms used for model training in their studies. On the other hand, although a large number of ML models have been utilized in ionospheric predictions, the performance of single models is limited by their respective shortcomings. For example, neural network (NN) performs well in learning complex patterns

from large quantities of available data, but often tends to easily overfit in the analysis of limited datasets (Hinton et al., 2012). Besides, due to the "black box" nature of NN, it is difficult to investigate potential relationships between inputs and inner structure of NN, so that NN outputs are often lacking of interpretability (Hastie et al., 2009). Comparatively, the bagging and boosting ML models like random forest (RF) and gradient boosting decision tree (GBDT) tend to show better performance and interpretability than NN on some small datasets (Zhukov et al., 2021; Han et al., 2022). To bridge this gap, the stacking





machine learning (SML) method is implemented to obtain better accuracy and generalization than a single model. SML uses the stacking strategy to leverage the efficiency of different models together, thus the possible errors of a single model can be complemented by other models. SML has been successfully applied to solve some ionospheric predictive problems. For example, Asamoah et al. (2024) proposed a stacked model combining three ML models to predict total electron content (TEC) over a single station, and demonstrated its better performance than the single models. Liu et al. (2024b) utilized a hybrid ensemble model to forecast ionospheric irregularities over Brazilian sector. However, to our knowledge, SML method has not been applied for EsI prediction till now.

Hence, in this article, we present an SML model for global ionospheric EsI prediction, where the physical observations are incorporated together with EsI derived from GNSS RO measurements. A variety of observations representing Es-related physical mechanisms are used as inputs of our model, including the vertical ion drift (VIC) driven by neutral wind shear, the GWs activity, and the solar and geomagnetic indices. The proposed SML method selects five widely adopted ML models, RF, light gradient boosting machine (LightGBM), eXtreme Gradient Boosting (XGBoost), support vector machine (SVM), and back propagation neural network (BPNN) as the base models, and a multilayer perceptron neural network (MLPNN) is utilized as the meta model to optimally integrate the predictions generated by base models. The prediction performance of SML is validated using RO-derived EsI and ionosonde data from different aspects, and is also compared with other prediction methods.

## 2 Data and Materials

### 2.1 GNSS RO-derived EsI

Constellation Observing System for the Meteorology, Ionosphere, and Climate (COSMIC) is a global RO mission launched in April 2006, with the goal of providing GNSS RO data for operational weather prediction, climate analysis, and space weather forecasting (Syndergaard et al., 2006). COSMIC consists of six LEO satellites with orbital altitude of 800 km and inclination of 72 °, and provides more than 2000 globally distributed TEC profiles per day during its full operation stage. COSMIC TEC profiles are the time series of the calibrated TEC below COSMIC LEO satellite altitude. Fig. 1 shows the traces of COSMIC TEC profiles on Aug 1, 2007. It indicates that COSMIC TEC data has a dense global coverage that is a great data source for global ionospheric investigations.



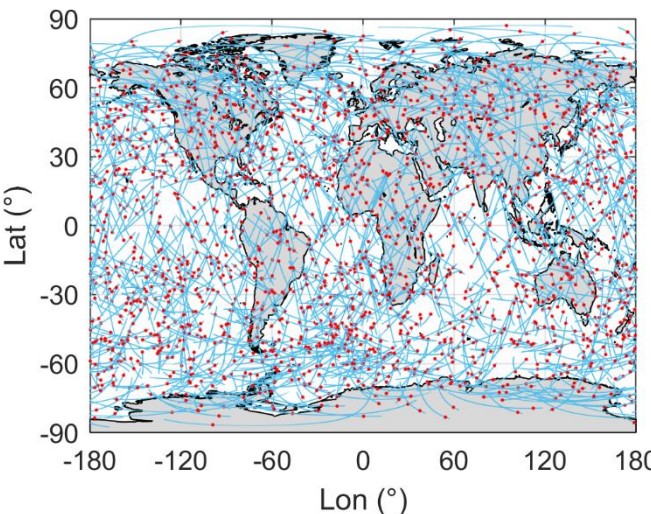

**Figure 1: The traces of COSMIC TEC profiles on Aug 1, 2007. The red points represent the locations corresponding to Smax.**

COSMIC TEC profiles have a high vertical resolution of better than 2 km, thus are suitable for the detection of Es
layers which are with small vertical scales. In this study, COSMIC TEC profiles during 2006–2019 are collected as the data
source for deriving EsI. Since the altitude range of Es layers considered in this paper is 90–130 km, a quality control process
is taken at first to remove the profiles with negative TEC values and the bottom heights higher than 90 km, then we use
single spectrum analysis (SSA) method as described in Hu et al. (2022) to obtain Smax index (unit: TECU km$^{-1}$), a proxy for
EsI, from qualified TEC profiles. The TEC disturbances caused by Es layers are extracted from original TEC profiles using
SSA. Smax is defined as the maximum vertical gradient of TEC disturbances in the altitude range of 90–130 km, and the
corresponding altitude is designated as the altitude of Es layer. The reasonableness and effectiveness of Smax as the proxy of
EsI have been verified by Niu et al. (2019) and Hu et al. (2022).

**2.2 Ionosonde EsI**

Es critical frequency (foEs) is a conventional parameter characterizing EsI. Ionosondes provide reliable ground-based local
foEs observations. In this study, foEs data downloaded from National Earth System Science Data Center of China (NESSDC)
and UK Solar System Data Center (UKSSDC) are used to validate the prediction results of SML. The foEs observations at 1-
hour intervals, all manually scaled, are obtained from 15 ionosondes. Fig. 2 shows the spatial distribution of ionosonde
stations.



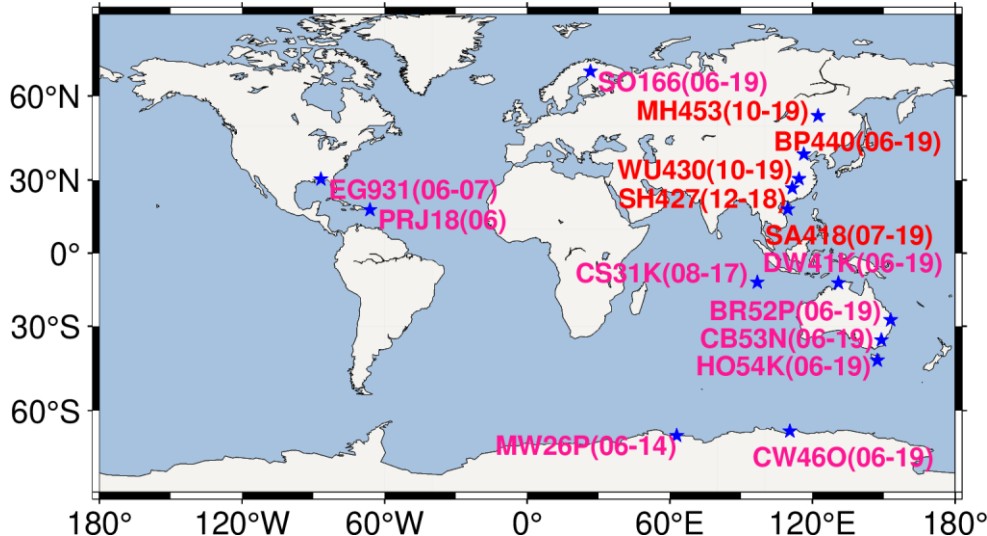

**Figure 2:** Spatial distribution of 15 ionosonde stations. The red and pink stations represent those from NESSDC and UKSSDC, respectively.

### 2.3 VIC simulated by HWM14

The wind shear theory has been proven to be the most significant factor influencing the formation of mid-latitude Es layers (Chu et al., 2014; Luo et al., 2021a). In the procedure of VIC driven by vertical wind shears in horizontal neutral winds, the metal ions are compressed into a thin layer by Lorentz force, and then the electrons drift along the magnetic field lines and converge to form an Es layer (Mathews 1998). We employ Horizontal Wind Model 2014 (HWM14) (Drob et al., 2015) to simulate VIC at the location of Es layer. The vertical ion drift velocity w caused by the horizontal neutral wind shear can be written as follows:

$$w = \frac{r \cos I}{1+r^2}U + \frac{\cos I \sin I}{1+r^2}V \tag{1}$$

where $U$ and $V$ are the zonal and meridional velocities of horizontal neutral wind, respectively, which are calculated by HWM14; $I$ is the geomagnetic inclination angle; $r = v_i/w_i$ is the ratio of the ion-neutral collision frequency to ion gyrofrequency (Nygrén et al. 2008), $w_i = eB/M$, $e$ and $M$ are the mass and charge of ion, respectively, and $B$ is the magnetic field strength. Qiu et al. (2019) and Yu et al. (2019) indicated that Es layers tend to appear in regions with negative vertical zonal wind shears ($\partial w/\partial z < 0$, where z is the altitude). Therefore, it is assumed that only positive values of VIC contribute to Es layer formation:



$$VIC = \begin{cases} -\dfrac{\partial w}{\partial z}, & -\dfrac{\partial w}{\partial z} > 0 \\ 0 \quad , & -\dfrac{\partial w}{\partial z} \le 0 \end{cases} \quad (2)$$

**2.4 GW activity extracted from GNSS RO data**

Recent studies have shown that the upward propagating GWs can transport energy and momentum from lower atmosphere to the mesosphere and lower thermosphere, causing ionospheric disturbances and contributing to the formation of Es layers (Qiu et al., 2023; Seid et al., 2023). COSMIC temperature profiles are able to cover a wide altitude range of 0–60 km with the vertical resolution better than 0.1 km, which are ideal data sources for studying GW activity. The proxy for GW activities, GW potential energy (Ep), can be calculated by following equations:

$$E_p = \frac{1}{2}\frac{g^2}{N^2}\left(\frac{T'}{\bar{T}}\right)^2 \quad (3)$$

$$N^2 = \frac{g}{\bar{T}}\left(\frac{\partial \bar{T}}{\partial z} + \frac{g}{c_p}\right) \quad (4)$$

where $g$ is the gravitational acceleration, N is the buoyancy frequency, $\bar{T}$ is the background temperature, $T' = T - \bar{T}$ is the temperature perturbation caused by GWs, and $c_p = 1004.5$ J $(\mathrm{kg} \cdot \mathrm{K})^{-1}$ is the isobaric heating capacity. To derive background temperature $\bar{T}$ that represents longer waves like tides and planetary waves, global daily temperature profiles are binned into

$10°\times15°$ latitude-longitude grids with an altitude interval of 0.1 km to obtain daily mean temperature maps at each altitude level, and S-transform is applied on each zonal component of mean temperature maps to derive gridded data of $\bar{T}$. The temperature perturbation $T'$ is obtained by subtracting $\bar{T}$ from original temperature profiles, and then GW Ep is calculated using Eq. (3) and (4). The detailed procedure for extracting GW Ep is described in Luo et al. (2021b).

**2.5 Solar and geomagnetic indices**

EsI is also affected by solar and geomagnetic conditions (Yu et al., 2021; Tang et al., 2022b). The solar radiation flux F10.7 and Dst indices are used to represent the solar and geomagnetic activity. We choose Dst index rather than 3-h Kp or Ap indices because Dst has a higher time resolution of 1 h. Fig. 3 shows the variations of F10.7 and Dst indices during 2006–2019.





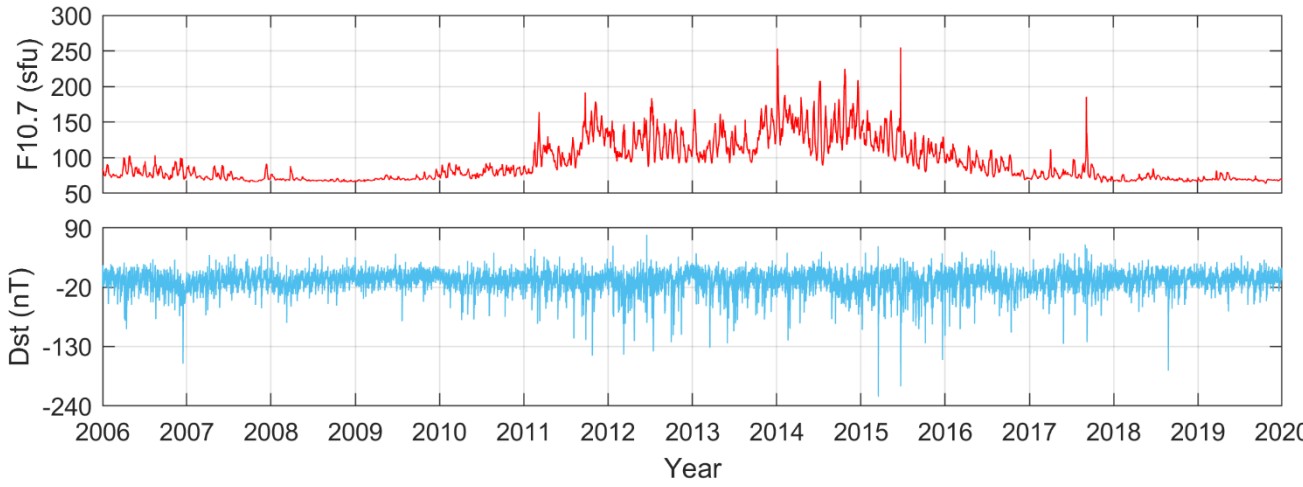

**Figure 3: Variations of F10.7 and Dst indices during 2006–2019.**

## 3 Methodology

### 3.1 Accuracy evaluation metrics

The mean error (ME), root mean square error (RMSE), and correlation coefficient (CC) are used as metrics to evaluate the accuracy of prediction results, which are calculated as:

$$
\begin{cases}
ME = \dfrac{1}{n}\sum_{i=1}^{n}\left(y_i - \hat{y}_i\right) \\[2mm]
RMSE = \sqrt{\dfrac{1}{n}\sum_{i=1}^{n}\left(y_i - \hat{y}_i\right)^2} \\[2mm]
CC = \dfrac{\mathrm{cov}\left(y_i,\ \hat{y}_i\right)}{\sigma_{y_i}\cdot\sigma_{\hat{y}_i}}
\end{cases}
\tag{5}
$$

where $n$ is the total number of prediction results; $y_i$ and $\hat{y}_i$ are the predicted and observed EsI, respectively; $\mathrm{cov}\left(y_i,\ \hat{y}_i\right)$ is the covariance between $y_i$ and $\hat{y}_i$; $\sigma_{y_i}$ and $\sigma_{\hat{y}_i}$ are the standard deviations of $y_i$ and $\hat{y}_i$, respectively. The units of both

ME and RMSE are TECU km$^{-1}$. CC has no unit.

### 3.2 Dataset configuration and segmentation

The proposed EsI prediction method aims to build a nonlinear functional model between the target (EsI) and inputs (spatiotemporal information and physical observations). Therefore, the time, latitude, longitude, and altitude corresponding





to each RO-derived EsI (Smax), as well as the VIC, GW Ep, F10.7, and Dst, are formed into samples and fed into SML. To reduce the input feature complexity and modeling costs, the time of each sample is expressed as follows:

$$Time = Year + (Doy + UT / 24) / 365.25 \tag{6}$$

where Doy and UT are day of year and universal time, respectively.

In SML method, the training and validation set is used for the training of the base models, and their prediction values become the training and validation set of the meta model. Since the cross-validation (CV) strategy is utilized for optimization of SML model (see Sect. 3.3.2), samples of the entire dataset collected during 2006–2019 are divided into two groups: training and validation set (80%, from 22 April 2006 to 31 December 2013), and testing set (20%, from 01 January 2014 to 31 December 2019). Note that there are fewer samples after 2014 due to the decline in the number of measurements caused by the aging and loss of COSMIC satellites. Nevertheless, it in turn allows for a longer time period of testing set and a more comprehensive evaluation of SML performance.

## 3.3 SML model development

### 3.3.1 ML models

SML combines the advantages from different ML models to obtain better performance than a single ML model. Diverse types of ML models should be selected to make SML fully incorporate their strengths, which require the selection of appropriate base models to simultaneously reduce the bias and variance. In this study, five ML models are utilized as base models, including RF, LightGBM, XGBoost, SVM, and BPNN. RF is a tree-based parallel ensemble ML algorithm using the bagging technique that is widely applied for classification and regression problems in GNSS and remote sensing tasks. RF is effective in reducing the variance of the model, and has an improved robustness to outliers. Comparatively, LightGBM and XGBoost are sequential ensemble ML models based on the boosting technique. They use the gradient boosting technique with outstanding performance in reducing bias of numerous datasets. SVM is an ML method based on the principle of structural risk minimization. It utilizes the structural risk minimization theory to suppress the overfitting problem and minimize empirical risk and confidence interval. BPNN is a widely used NN model with high adaptability and learning ability for regression problems. It comprises three types of fully connected layers, i.e., the input layer, hidden layer(s), and output layer, which help to better capture complex nonlinear relationships.

Furthermore, MLPNN is also a common NN model to solve regression problems. It has a similar structure with BPNN, while the main difference between MLPNN and BPNN lies in their activation functions. Here we use MLPNN as the meta model to find the optimal combination of base models, and the structures of the base models and meta model are shown in Fig. 4.







**Figure 4: The structures of RF, LightGBM, XGBoost, SVM, BPNN, and MLPNN.**

The mathematical expressions for all the ML models used are presented in Eq. (7)–(12):

$$RF(x) = \frac{1}{M}\sum_{m=1}^{M} T_m(x) \tag{7}$$

where $x$ is the inputs, $M$ is the number of trees, and $T_m(x)$ is the $m$th tree output.

$$LightGBM(x) = \frac{1}{M}\sum_{m=1}^{M} T_m(x)W_m \tag{8}$$

where $W_m$ is the weight of the $m$th tree. LightGBM is similar to GBDT, but compared with the depth-wise tree growth approach, it grows trees using the leaf-wise approach that focuses on nodes with the highest loss change, which is better at handling large datasets and improving prediction accuracy.



$$XGBoost(x) = \frac{1}{M} \sum_{m=1}^{M} T_m(x) W_m \tag{9}$$

XGBoost is also similar to GBDT, while it offers a parallel tree boosting algorithm to improve computation efficiency. Actually, LightGBM and XGBoost are new optimized implementations for GBDT using different techniques.

$$SVM(x) = w\varphi(x) + b$$
$$s.t. \quad \min\left[\frac{1}{2}\|w\|^2 + C \sum_{i,j=1}^{L} (\xi_i, \xi_j)\right] \tag{10}$$

where $w$ is the weight vector, $\varphi$ is the nonlinear mapping function, $b$ is the bias, $L$ is the number of input samples, $C$ is the penalty factor specifying the degree of penalty for outliers, and $\xi_i$ and $\xi_j$ are relaxation factors. Equation (9) can be solved

by introducing Lagrange multipliers to obtain the regression function of SVM, in which $\varphi$ is usually replaced by the radial basis function kernel $K(x_i, x_j) = \exp(-\gamma \|x_i - x_j\|^2)$, where $\gamma$ is the kernel parameter. The details of SVM algorithm can be found in Yetilmezsoy (2019).

$$BPNN_k(x) = f\left(\sum_{j=1}^{L} w_{kj} x_j + b_k\right)$$
$$MLPNN_k(x) = g\left(\sum_{j=1}^{L} w_{kj} x_j + b_k\right) \tag{11}$$

where $BPNN_k$ and $MLPNN_k$ are the outputs of the kth neuron, $f(\cdot)$ and $g(\cdot)$ are the activation functions of BPNN and

MLPNN, respectively. We select the sigmoid and hyperbolic tangent functions as the activation functions of BPNN and MLPNN, respectively, which can be written as:

$$f(x) = \frac{1}{1 + e^{-x}}$$
$$g(x) = \frac{2}{1 + e^{-2x}} - 1 \tag{12}$$

### 3.3.2 Model optimization

Hyperparameters are the internal configuration parameters for ML models. The optimization of hyperparameters is important

for improving the accuracy and generalizability of ML models. To determine the optimal hyperparameters while maintaining a relatively low computational cost, the grid search method is adopted to optimize the two hyperparameters with the greatest impact on the model. Specifically, for each model, a set of candidate values of the two hyperparameters to be optimized is defined in the parameter space, the model performance for each hyperparameter combination is evaluated, and the best performing hyperparameter combination is defined as the optimal hyperparameters. During the grid search process, a five-

fold CV is utilized. The training and validation set is randomly divided into five non-overlapping folds. For each iteration of





training and evaluation, the *i*th fold ($i = 1, 2, \ldots, 5$) is used as the validation set, and the remaining folds are used as the training set. The average results of these iterations are denoted as the final performance evaluation.

Figure 5 shows the optimized hyperparameters, the candidate values, and the optimal values (denoted by red asterisks) of hyperparameters for each ML model. For RF (bagging model), the number of leaf nodes and the number of trees

are selected as the hyperparameters to be optimized, and they are optimally determined as 200 and 200, respectively. While for LightGBM and XGBoost (boosting model), the number of leaves and the maximum depth play the more important roles in improving model performance, and they are optimally computed to 63 and 9, respectively. As shown in Eq. (9), the penalty factor C and the kernel parameter γ have significant impacts on the performance of SVM regressor. In this sense, they are selected for optimization with optimal values of 10 and 1, respectively. For BPNN and MLPNN, the number of

hidden layer(s) and the neuron number in each hidden layer are key hyperparameters in determining the accuracy of the network. Since one hidden layer-based NN can approximate the arbitrarily small error among most bounded continuous functions (Hornik et al., 1989), we only use one hidden layer in BPNN, while two hidden layers are adopted in MLPNN to better combine the predictions from base models. The optimal neuron number in hidden layer(s) can be determined empirically based on the range from $2\sqrt{n} + \mu$ to $2n + 1$, where n and μ are the neuron numbers in input and output layers,

respectively. Therefore, the neuron numbers of BPNN and MLPNN are validated from 6 to 17 and from 4 to 11, respectively, and the optimal neuron numbers corresponding to the minimum RMSE are 16 for BPNN, 6 and 7 for MLPNN, respectively. By optimizing these ML models, their performance, generalizability and interpretability are improved so that they are more suitable for the specific task of EsI prediction in this study.



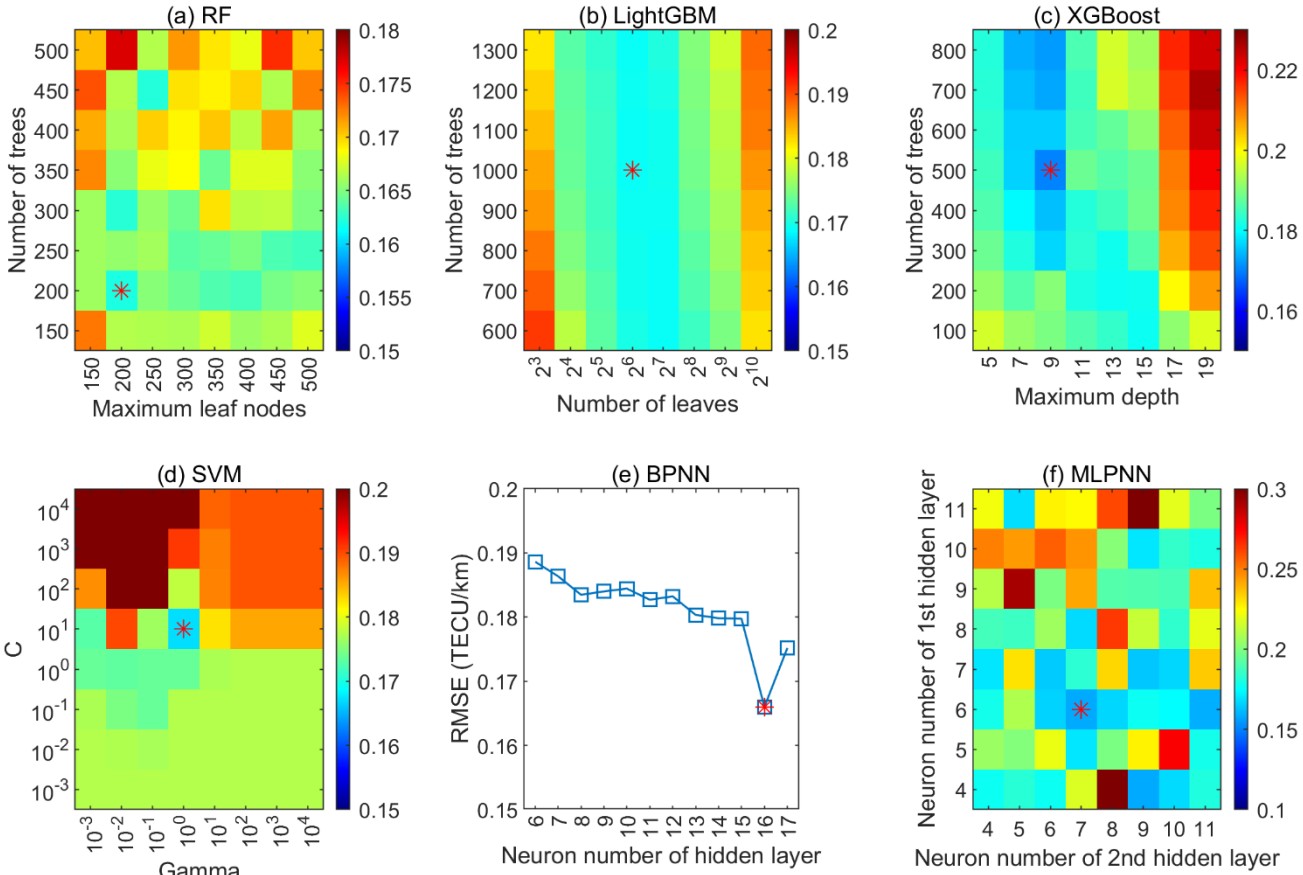

**Figure 5: Grid search results (RMSE) for the optimization of RF, LightGBM, XGBoost, SVM, BPNN, and MLPNN. The red asterisks denote the optimal hyperparameter combination of each ML model.**

### 3.3.3 SML model architecture

The stacking strategy, proposed by Wolpert (1992), is aimed to improve the base models by incorporating the outputs of multiple base models into the training process of the meta model. Compared with other ensemble method like bagging and boosting, it has better ability to reduce both variance and bias. In the training stage of SML model, the outputs of RF, LightGBM, XGBoost, SVM, and BPNN on the training and validation set are fed into MLPNN as its input data, and its outputs are the final predicted EsI. This process is also similar in the test stage. The framework of SML is shown in Fig. 6.



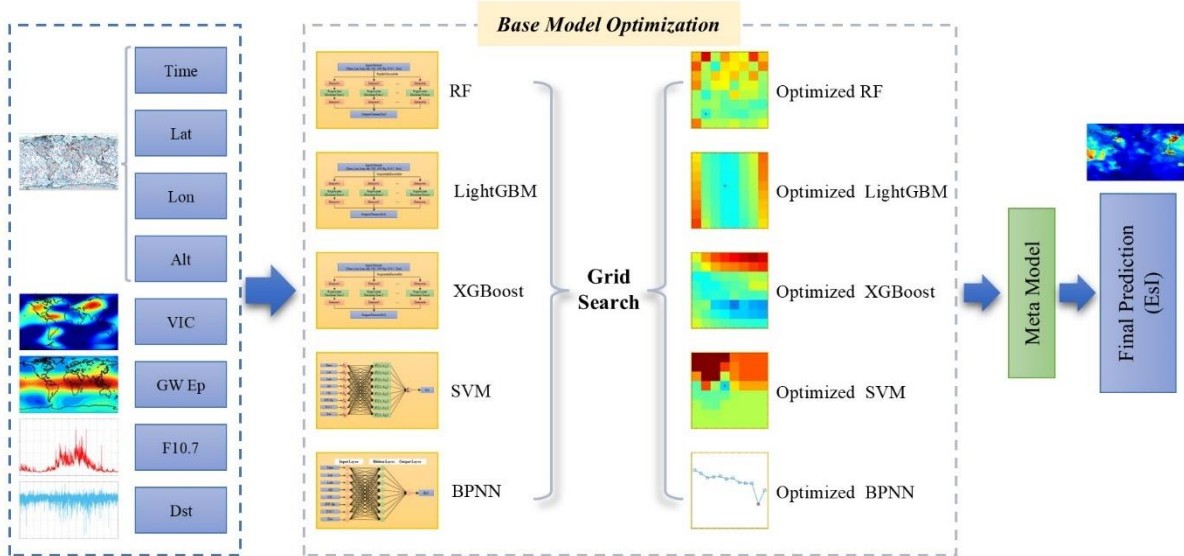

**Figure 6: Framework of the SML model.**

## 4 Results

### 4.1 Comparison of SML model and base models

Both the SML model and the base models with the optimal hyperparameters are fitted on the training set. Then they are employed to make predictions on the testing set, and the prediction results are compared with the ground truth. Fig. 7 illustrates the histograms and the density scatter plots of the comparisons of EsI predicted by SML and base models with ground truth. SML shows the much more aggregated histogram than those of base models, which means that SML has the best agreement with ground truth among all ML models, with the minimum ME/RMSE of 0.032/0.158 TECU km$^{-1}$.

Compared to the maximum ME and RMSE of 0.053 and 0.170 TECU km$^{-1}$ for the base models, SML has the improvement of 39.6% and 7.1%, respectively. The density scatter plots show that SML also has the highest CC of 0.891. As mentioned above, RF is more robust for outliers and more effective in reducing variances, thus has a lower RMSE than other base models. Comparatively, the other base models, especially BPNN, play an important role in reducing bias, and they outperform RF in terms of the overall prediction accuracy of EsI, as demonstrated by their MEs. By combining the strengths

of different types of ML models, SML is able to achieve predictions with both lower biases and lower variances compared with all base models.



**Figure 7: (a) Histograms and (b) density scatter plots of the comparisons of EsI predicted by SML and by base models with ground truth.**

In addition to the overall performance, the global distributions of EsI predicted by the models are also compared. Figure 8 presents the latitude-longitude maps of the differences between ground truth and the EsI predicted by SML and by base models on the testing set. Here the spatial resolution of EsI maps is set as $2.5° \times 5°$. Specifically, RF has larger deviations than other models in North America (30 N–50 N and 80 W–120 W), South Atlantic Anomaly (SAA) zone




(30 °S–60 °S and 20 °W–40 °E), and near the geomagnetic equator (denoted by the red line in Fig. 8(a)), which are the regions with smaller EsI due to the near horizontal geomagnetic field lines (Yu et al., 2019; Luo et al., 2021a). RF also has a considerable underestimation of EsI in the Arctic region. The other base models, especially BPNN, have overall smaller biases than RF, while they have more outliers exhibited as localized small patches with larger prediction errors at mid- and

low-latitudes. On the other hand, SML has the best prediction accuracies and the least number of outliers in the regions mentioned above, which is due to that SML avoids the shortcomings of different base models by selectively integrating their outputs.

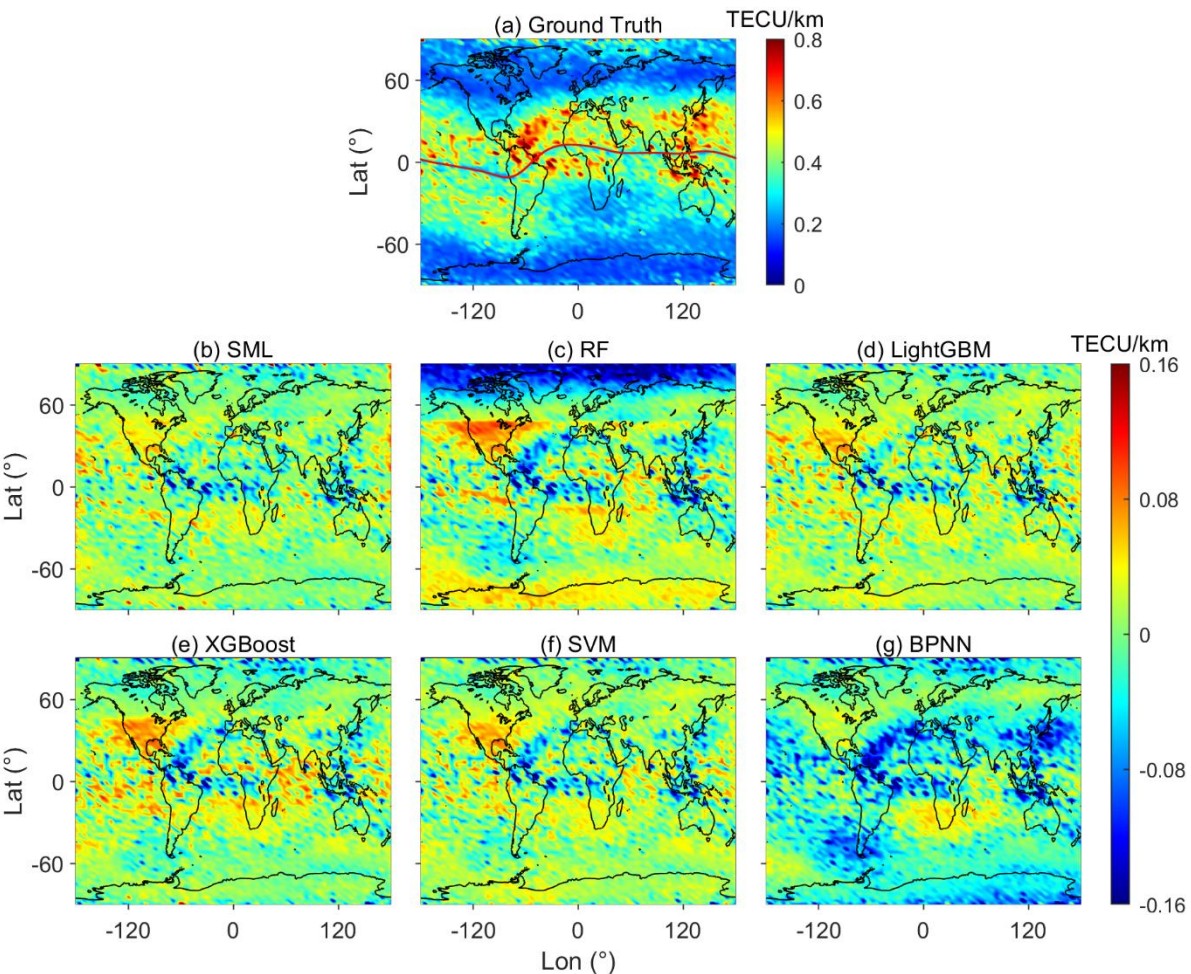

**Figure 8: Latitude-longitude distribution of (a) ground truth, and the difference between ground truth and the EsI predicted by (b) SML, (c) RF, (d) LightGBM, (e) XGBoost, (f) SVM, and (g) BPNN on the testing set. The red line denotes the geomagnetic equator.**





Based on the above comparisons, SML has the highest CC and the smallest ME and RMSE, i.e., the best prediction performance. In the following sections, only SML model is selected to assess the ability in reconstructing the complex long-term and short-term characteristics of EsI morphology, and we also compare the SML predictions with ionosonde observations for external validation of the prediction performance.

## 4.2 Long-term evaluation of SML performance

The long-term evaluation of SML prediction performance is conducted on the whole testing set, i.e., 2014–2019. Figure 9 presents the latitude-longitude and latitudinal distributions of ground truth, SML-predicted EsI, and the corresponding error maps in four seasons, which are categorized as MAM (March, April, and May), JJA (June, July, and August), SON (September, October, and November), and DJF (December, January, and February). Visual inspection shows that SML accurately simulate the seasonal variation of EsI. SML successfully shows the larger EsI that peaks in the banded area at mid-latitudes of summer hemisphere and reaches the valley values in winter hemisphere, which is primarily dominated by the seasonal variation of meteor flux and the resulting metallic ion content, coupled with the neutral wind shear (Haldoupis et al., 2007). The weaker EsI in North America, SAA zone, and along the geomagnetic equator due to the lower geomagnetic inclination angle is well reconstructed by SML predictions. Furthermore, the SML-predicted latitudinal distribution of EsI also agrees well with ground truth in all the four seasons. The larger EsI moves northward or southward with seasonal variations, which is under the control of wind shear at mid-latitudes. While at latitudes higher than 70 °, there is also strong EsI which is larger than that at 60 °, and this is no longer due to the wind shear but the vertical transport of ions and electrons caused by GWs propagating upward along near vertical geomagnetic field lines (Kirkwood and Nilsson, 2000). The clearly reconstruction of these two larger EsI with different reasons indicates that SML model can simultaneously consider different physical mechanisms of Es layer formation. The SML prediction errors are between ±0.1 TECU km$^{-1}$ in most areas, demonstrating the excellent prediction performance in long-term EsI prediction. The ME of SML predictions for the four seasons are 0.004/-0.005/0.001/0.012 TECU km$^{-1}$, and RMSE are 0.146/0.166/0.143/0.176 TECU km$^{-1}$, respectively. Nevertheless, we can see from the error maps that the areas of underestimation beyond ±0.1 TECU km-1 are mainly concentrated at the peak of EsI in the summer hemisphere. The possible explanation for this phenomenon is that larger EsI is the minority of the training set (predominantly occurs in the summer hemisphere only), and the trained model fits this part of data slightly worse than smaller EsI. Improvement in the prediction performance for larger EsI should be required for future study.







**Figure 9: Latitude-longitude and latitudinal distributions of ground truth, SML-predicted EsI, and the corresponding error maps in four seasons.**

Figure 10 shows the local time (LT)-day of year (DOY) distributions of ground truth and SML-predicted EsI at different latitude ranges in 2014–2019. Larger EsI mainly exists in daytime, increasing after sunrise and decreasing after sunset. The EsI tidal signatures reconstructed by SML, mainly dominated by the wind shear and atmospheric tides (Yu et al., 2019), is consistent with ground truth, which can be identified in summer days with diurnal tides (start around 10 LT) occurring at low-latitudes (30°N–30°S) and semi-diurnal tides (start around 8 LT and 16 LT) occurring at mid-latitudes (30°N–60°N and 30°S–60°S). Although the EsI seasonal variation in the Southern Hemisphere (SH) is opposite to that in the



Northern Hemisphere (NH), the diurnal and semi-diurnal tides can still be discerned, only with a slightly lower peak intensity. Figure 10 indicates the effectiveness of SML in reconstructing tidal signatures of EsI.

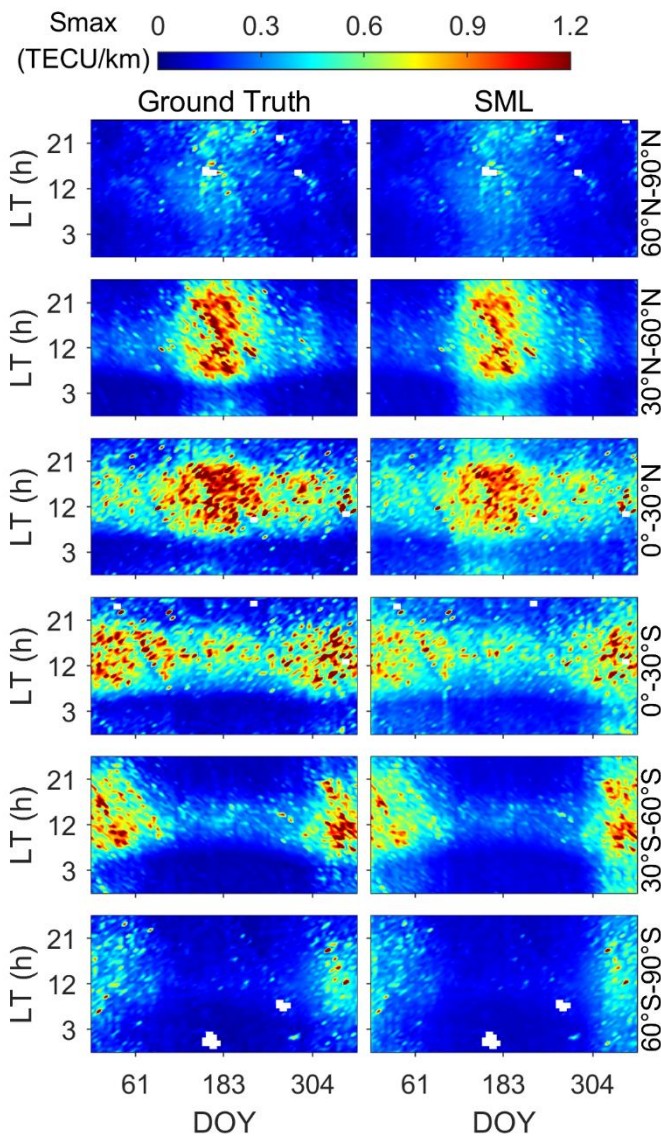

**Figure 10: LT-DOY distributions of ground truth and SML-predicted EsI. The right label of the right panel of each row represents the latitude range of this row.**

Figure 11 plots the latitudinal distribution of daily SML-predicted EsI and the daily RMSE on the whole testing set, with blank areas indicating the days without EsI data. The results present that the morphology characteristics of SML

predictions are close to those of ground truth. SML succeeds in capturing the hemispheric asymmetry of EsI, i.e., EsI is



generally slightly higher in the NH summer than in the SH summer of the same year, which is also found by Luo et al. (2021a) and Xu et al. (2022). This is mainly due to the lower EsI in SAA zone caused by the distribution of horizontal geomagnetic field, which diminishes the EsI over the corresponding latitude zones. Furthermore, the daily RMSE in Fig. 11 is generally stable below 0.2 TECU km$^{-1}$, with unusual sudden enhancements only in a few days. Overall, these results fully
demonstrate the good ability of SML for stable EsI prediction and characteristics reconstruction in long-term periods.

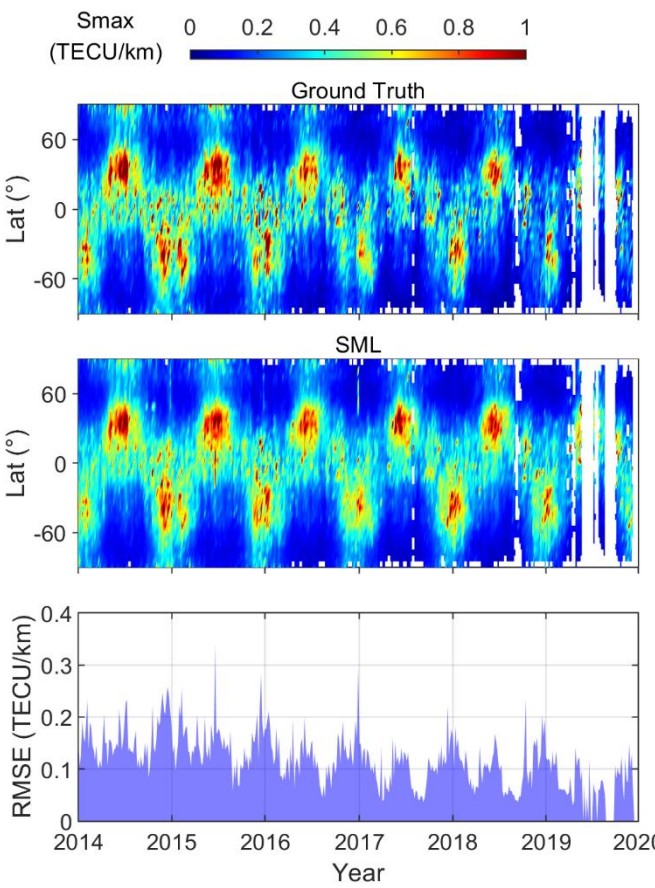

**Figure 11: Latitudinal distribution of daily ground truth and SML-predicted EsI, as well as the daily RMSE on the testing set.**

**4.3 Short-term evaluation of SML performance**

The response of EsI to geomagnetic storms has been widely reported (Resende et al., 2020; Moro et al., 2022; Tang et al., 2022b), which is usually a combined effect of neutral wind and electric field variations. We have conducted two case studies to evaluate the short-term prediction performance of SML during the geomagnetic quiet and storm time periods. Figure 12





shows the latitude-longitude and latitudinal distributions of ground truth, SML-predicted EsI, and the corresponding error

maps on two quiet days, 04 Jul 2014 and 24 Jan 2018. Compared with ground truth, SML can effectively predict the general distribution of EsI, particularly the considerable agreement in latitudinal distribution, which is similar to that in Fig. 9. The prediction errors are mostly within ±0.1 TECU km$^{-1}$, with a small number of underestimates which are with larger errors mainly existing in the summer hemisphere. The ME/RMSE for the two quiet days are -0.006/0.183 TECU km$^{-1}$ and 0.006/0.133 TECU km$^{-1}$, respectively.


**Figure 12: Latitude-longitude and latitudinal distributions of ground truth, SML-predicted EsI, and the corresponding error maps on 04 Jul 2014 and 24 Jan 2018.**

Figure 13 shows the latitude-longitude and latitudinal distributions of ground truth, SML-predicted EsI, and the corresponding error maps on two storm days, 07 Dec 2014 (moderate storm, Dst = -43 nT) and 22 Jun 2015 (major storm, Dst = -121 nT). Compared to quiet days, the EsI distributions during storm days are more complex, showing more irregular patches of EsI enhancement. The general distributions of SML predictions still agree well with ground truth, while there are more outliers in the summer hemisphere and at low latitudes compared to quiet days, as shown in the error maps. The

ME/RMSE for the two storm days are 0.008/-0.004 TECU km$^{-1}$ and 0.278/0.268 TECU km$^{-1}$, respectively. In addition, SML has more overestimations of EsI on 07 Dec 2014, while presents both overestimations and underestimations on 22 June 2015. Liu et al. (2022) reported that EsI usually has a decrease during moderate storms than during quiet time, and presents a



complex variation during major storms. Although their study is from a climatological perspective, it may explain our prediction results during geomagnetic storms. Nonetheless, Figs. 12 and 13 suggest that SML model has a reliable ability for
short-term EsI prediction under different geomagnetic levels.

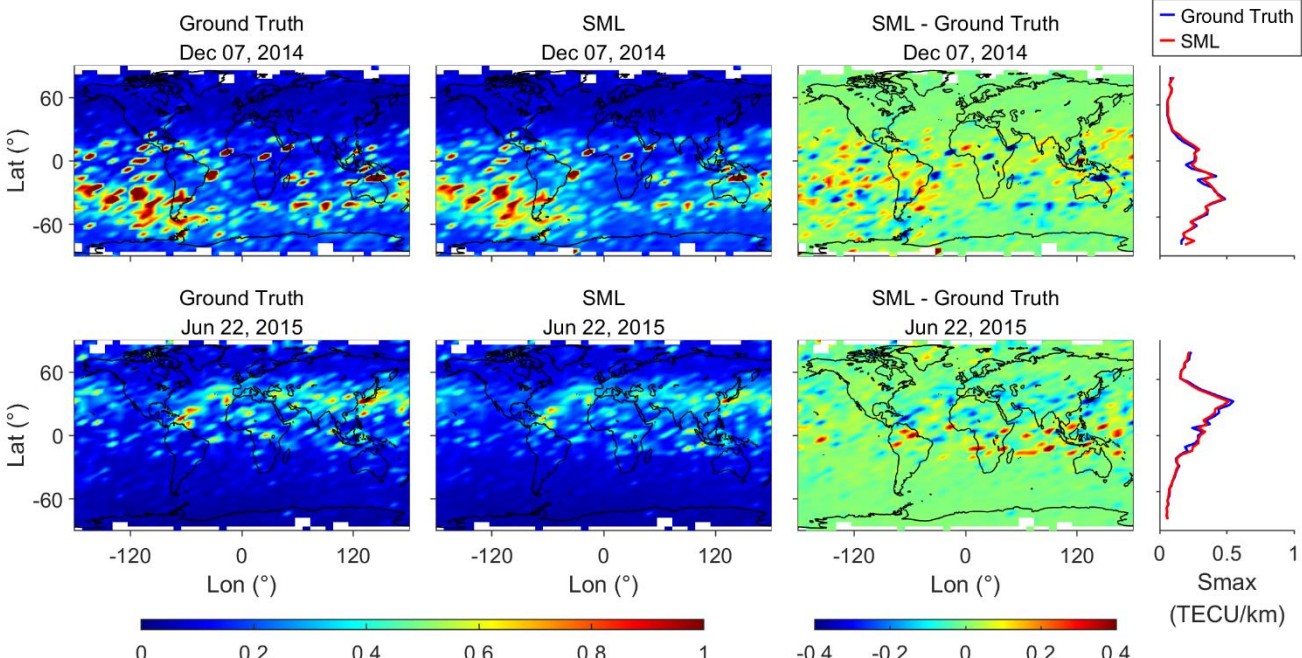

**Figure 13: Latitude-longitude and latitudinal distributions of ground truth, SML-predicted EsI, and the corresponding error maps on 07 Dec 2014 and 22 Jun 2015.**


## 4.4 External validation using ionosonde observations

In the previous empirical modeling of EsI, ionosonde foEs observations are usually used to verify the accuracy of RO measurements (Niu et al., 2019; Hu et al., 2022). To compare the SML-predicted EsI (Smax) and ionosonde foEs on the testing set, they should be matched under a specific spatiotemporal window. Luo et al. (2019) indicated that the influence of
the increase in the spatial window on the matching results is much greater than that in the time window. Therefore, we adopt the window of (0.5°, 0.5°, 1 h) to ensure both the amount and consistency of the matched pairs. Fig 14 demonstrates the scatter plots of the matched ground truth and the SML-predicted EsI with ionosonde foEs. The results show that the fitted equation between SML predictions and foEs is much close to that between ground truth and foEs. The fitted RMSE of 0.122 TECU km$^{-1}$ for SML is only slightly worse than that of the ground truth of 0.121 TECU km$^{-1}$, while the CC becomes even





better, from 0.716 for ground truth to 0.727 for SML. The high consistency between the metrics of SML predictions and ground truth indicates the good performance of SML.

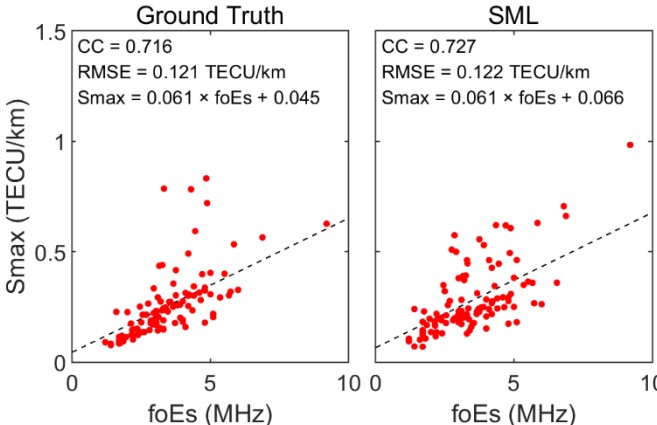

**Figure 14: Scatter plots of the matched ground truth and the SML-predicted EsI with ionosonde foEs.**


Furthermore, we verify the consistency of the long-term trends of SML results and ionosonde observations. Three ionosondes located at different latitudes, WU430, BP440, and MH453, are selected for evaluation. Fig. 15 shows the daily maximum of the SML-predicted EsI and foEs over the selected ionosondes during 2014–2019. The climatological variations of SML-predicted EsI correspond well with those of the ionosonde foEs, and both of them have the same decreasing trend 400 from south to north. The CCs of SML-predicted EsI and ionosonde foEs over the three ionosondes are 0.645, 0.739, and 0.684, respectively.



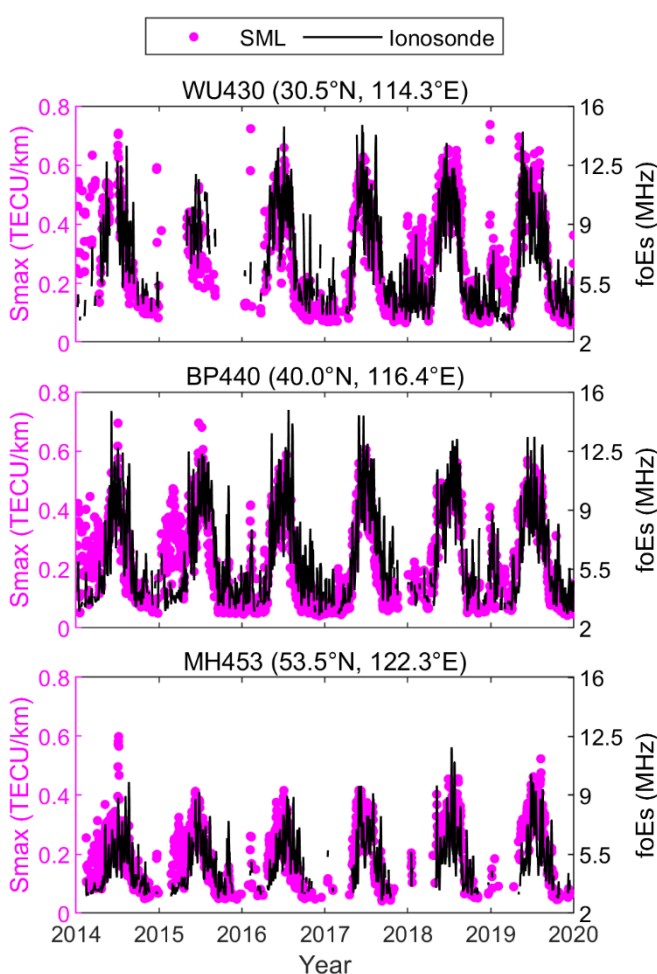

**Figure 15: Time series of daily maximum of SML-predicted EsI and foEs over WU430, BP440, and MH453 ionosondes during 2014–2019.**

# 5 Discussion

## 5.1 Advantage of incorporating physical observations in EsI prediction

To investigate the effect of incorporating physical observations on the prediction performance of SML model, we have compared four combinations of different input variables, including 1. EsI, Time, Lat, Lon, Alt, VIC, GW Ep, F10.7, Dst; 2. EsI, Time, Lat, Lon, Alt, VIC, GW Ep; 3. EsI, Time, Lat, Lon, Alt, F10.7, Dst; and 4. EsI, Time, Lat, Lon, Alt. Table 1 represents the metrics of SML with four input variable combinations on the testing set. The SML model with Combination 1 has the smallest ME and RMSE compared to other combinations. The models with Combinations 2 and 3 have the



ME/RMSE of 0.038/0.162 TECU km$^{-1}$ and 0.045/0.179 TECU km$^{-1}$, which are also smaller than those of Combination 4.
Furthermore, the monthly RMSE of SML with different input variables are shown in Fig. 16. It is evident that the monthly RMSE of SML with Combination 4 are larger than the SML models with other combinations during March–November. The SML with Combination 1 performs better than other models during January–November. These results show the necessity of incorporating multiple related physical factors to consider the interactions of different atmospheric layers as a coupling system when constructing the Es prediction model.


**Table 1. Metrics of SML with different input variables**

| Input Variables | ME (TECU km$^{-1}$) | RMSE (TECU km$^{-1}$) |
|---|---|---|
| EsI, Time, Lat, Lon, Alt, VIC, GW Ep, F10.7, Dst | 0.032 | 0.158 |
| EsI, Time, Lat, Lon, Alt, VIC, GW Ep | 0.038 | 0.162 |
| EsI, Time, Lat, Lon, Alt, F10.7, Dst | 0.045 | 0.179 |
| EsI, Time, Lat, Lon, Alt | 0.049 | 0.184 |

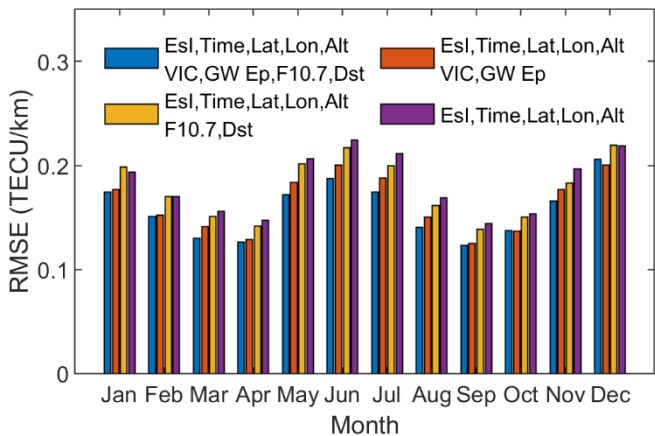

**Figure 16: Monthly RMSEs of SML with different input variables.**


In recent years, some methods have been proposed for EsI modeling and prediction. Niu and Fang (2023) used COSMIC RO data to develop an empirical model that reproduces the climatological characteristics of EsI at low- and mid-latitudes with averaged deviation of 0.23 MHz. Emmons et al. (2023) presented two improved prediction model for EsI and demonstrated better performance than those of the empirical models. Although the above methods achieved considerable EsI
prediction performance, the lack of Es-related physical observations limited the further improvements of model accuracy.



Tian et al. (2023) conducted the importance ranking of potential Es-related lower atmospheric parameters, based on which they selected the input variables for their prediction model, but they did not consider VIC, the most important physical factor. In this study, we comprehensively incorporate the physical observations that have been proven to have significant correlations with EsI. Hence, we have obtained a better performance than the previous models with only the EsI information as inputs.

## 5.2 Comparison of SML and other EsI estimation models

We have collected EsI estimation models using RO measurements from literatures in recent years. These models utilize different COSMIC RO products to derive various EsI proxies using statistical or ML methods. For intuitive comparison, all EsI predictions are validated by ionosonde foEs observations on the same testing set using the same collocation window in Sect. 4.4, and their metric units are all converted to MHz. The comparison results are listed in Table 2. SML has the best prediction performance, and its RMSE is considerably smaller than all the other four models, with the improvements of 40.5%, 35.5%, 33.5%, and 20.1%, respectively. Emmons et al. (2023) used an ML model (SVM regression) to achieve a smaller RMSE than the other three empirical models, while it is still larger than the SML RMSE of this work. This may be due to that the single SVM model is not as robust as SML.

Table 2. Comparison results of EsI estimation models.

|  | COSMIC product | EsI proxy | Method | RMSE (MHz) |
|---|---|---|---|---|
| Yu et al. (2022) | S4 index | S4max | Nonlinear least-squares fitting | 1.787 |
| Niu and Fang (2023) | TEC profiles | Smax | Multivariable functional fitting | 1.650 |
| Liu et al. (2024a) | Electron density profiles | Electron density | Linear fitting | 1.601 |
| Emmons et al. (2023) | 50 Hz SNR profiles | Normalized SNR | SVM regression | 1.331 |
| This paper | TEC profiles | Smax | SML | 1.064 |





## 6 Conclusions

This study proposes an SML method for global EsI prediction, in which a variety of Es-related physical observations are

incorporated as inputs together with EsI derived from GNSS RO measurements. SML combines the strengths of the optimized base models to obtain lower prediction bias and variance. Taking RO-derived EsI as reference, the ME and RMSE of SML are 0.032 TECU km$^{-1}$ and 0.158 TECU km$^{-1}$, respectively, and the reductions compared with the maximum ME and RMSE of base models are 39.6% and 7.1%, respectively. The evaluation results during 2014–2019 show that SML performs well in the prediction and the characteristics reconstruction of both long-term and short-term EsI variations. Taking

ionosonde foEs observations as reference, SML shows better performance in EsI prediction compared to the existing methods, with the improvements of 20.1%–40.5%. Overall, this study presents an effective tool for high-precision global EsI prediction, which can be expected to provide valuable information for ionospheric irregularities monitoring and space weather forecasting. The method's incorporation of multiple Es-related physical factors is of significant contribution for deepening the understanding of complex interactions between lower atmosphere, thermosphere, ionosphere, and solar-

terrestrial environment.

## Author contributions

TH: Conceptualization, Methodology, Software, Validation, Formal analysis, Writing – original draft, Visualization. XX: Investigation, Writing – original draft, Writing – review & editing, Supervision, Funding acquisition. JL: Data curation, Writing – review & editing, Funding acquisition. JH: Methodology, Data curation. HL: Software, Formal analysis.

## Competing interests

The contact author has declared that none of the authors has any competing interests.

## Data availability

The COSMIC RO data are available from COSMIC Data Analysis and Archive Center (CDAAC) at University Corporation for Atmospheric Research (UCAR) at https://data.cosmic.ucar.edu/gnss-ro/cosmic1/ (UCAR COSMIC Program, 2022). The

ionosonde data are available from National Earth System Science Data Center of China (NESSDC) at http://wdc.geophys.ac.cn/dbList.asp?dType=IonoPublish (National Earth System Science Data Centre, 2025) and UK Solar System Data Center (UKSSDC) at https://www.ukssdc.ac.uk/wdcc1/ionosondes/secure/iono_data.shtml (UK Solar System Data Centre, 2025). The F10.7 and Dst indices are available from NASA/Goddard Space Flight Center's OMNIweb at https://omniweb.gsfc.nasa.gov/form/dx1.html (Papitashvili and King, 2025).



**Acknowledgements**

We kindly acknowledge the COSMIC Data Analysis and Archive Center (CDAAC) at University Corporation for Atmospheric Research (UCAR) for providing the COSMIC RO data, and NASA/Goddard Space Flight Center's OMNIweb for providing the F10.7 and Dst indices. The authors also acknowledge the use of HWM14 provided by National Space Science Data Center (NSSDC).

**Financial Support**

This study is supported by the National Natural Science Foundation of China under Grant Nos. 42074027, 42174017, 41774033, and 41774032.

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
