# Peer review of "Global ionospheric sporadic E intensity prediction from GNSS RO using a novel stacking machine learning method incorporated with physical observations"

_EGUsphere, 2025_

## Author Comment (AC1)

**Response to Reviewers**

**Directory**

**Reply to Reviewer #1 Dr. Yosuke Yamazaki**

Paper Title: "Global ionospheric sporadic E intensity prediction from GNSS RO using a novel stacking machine learning method incorporated with physical observations".

Authors: T. Hu, X. Xu, J. Luo, J. Hou, H. Liu

Preprint: egusphere-2025-1549

**Summary comment**

This is a well-written and insightful paper on machine learning modeling of sporadic E (Es) layer intensity. The study is particularly novel in two key aspects: (1) the application of a stacking machine learning (SML) approach, and (2) the integration of atmospheric parameters as model inputs. The authors convincingly demonstrate that their model not only captures the salient features of Es climatology but also outperforms existing models. The topic and overall quality of the paper are well aligned with the journal's scope. Below are my comments on the manuscript. Most are minor, though a few may be considered major.

**Reply:** Thanks for your kindly evaluation on our manuscript, as well as your constructive suggestions and detailed comments. We revised the manuscript according to your recommendations, and the revisions also took into consideration the comments from another reviewer.

Two important revisions are made following your comments. First, we modified the abstract by adding the statement of the two novelties you mentioned. We stated the basic principle of SML method, and included the specific physical observations as model inputs in the abstract. Second, we made use of two other ionosonde stations, DW41K (low latitude) and SO166 (high latitude) to replace WU430 and MH453 in Figure 15, so that Figure 15 can comprehensively show the evaluation results in all of low, mid and high latitudes. Third, in Section 5.1, we modified the combinations of input variables for the evaluation by removing them one by one. The sequence of removal is Dst, F10.7, GW Ep and VIC. We found that the increases of ME and RMSE after each removal, which represent the contribution of the corresponding parameter, are larger for VIC and F10.7, and smaller for GW Ep and Dst. We listed some possible reasons for explanation, and we also added the summary at the end of the corresponding paragraph.

In this reviewer response letter, your remarks are responded point-by-point. In the revision, all the changes made on the original manuscript have been highlighted in red color.

**Detailed Comments**

(1) Abstract

The abstract does not clearly highlight the novelty of the study, as it omits key details that are essential for understanding the significance of the work. The authors should briefly state what the SML method is. I found this information later in the main text.

"SML combines the advantages from different ML models to obtain better performance than a single ML model."

Also, the use of vertical ion convergence (VIC) and gravity wave (GW) potential energy as model inputs should be explicitly stated. The present abstract only mentions "physical observations" without explaining what they are.

**Reply:** We greatly appreciated your constructive suggestion about the novelty of our manuscript. We modified the abstract by adding the statement of the two novelties you mentioned. We stated the basic principle of SML method, and included the specific physical observations as model inputs in the abstract. (Lines 14–16)

(2) (l. 33) "vertical motion of gravity waves"

This sounds strange. Do the authors mean "vertical air motion due to gravity waves"?

**Reply:** Thanks. Here we mean "upward propagation of gravity waves".

(3) (l. 35) "Therefore, the high precision modeling and ..."

This "Therefore" sounds out of place. The authors did not explain why "the high-precision modeling and prediction of Es layers" are crucial for space weather forecasting.

**Reply:** Thanks to your detailed comment. To explain this statement, we added the description about the potential impact of Es layers on radio communication and satellite navigation in Lines 36–37 as "Es layers may cause scintillations on the propagations of radio signals and severely affect the radio communication and satellite navigation".

(4) W (l. 48) "physical observations"

It is unclear what they are. Please be specific.

**Reply:** Thanks for pointing this issue out. We listed the physical observations that should be considered following the sentence. (Lines 51–52)

(5) (l. 57) "there were little data on Es-related physical mechanisms"

What are considered as "data on Es-related physical mechanisms"? Please be specific.

**Reply:** Thanks for the comment. We added that data about neutral wind, gravity waves, solar and geomagnetic activity can be considered as "data on Es-related physical mechanisms", and they should contribute to better performance of Es layer reconstruction and prediction. (Lines 61–62)

(6) (l. 57) "On the other hand"

This sounds out of place.

**Reply:** Thanks. Here we change our discussion from incorporation of physical data to the shortcomings of single ML models. To properly represent the transition, we replaced "on the other hand" with "in addition". (Line 63)

(7) (l. 65) "better accuracy and generalization"

I understand "better accuracy" but not "better generalization". What does the latter mean?

**Reply:** "Generalization" in artificial intelligence field refers to a model's ability to make accurate predictions on new, previously unseen data. That means, a model with good generalization can learn features from training data and apply them to new samples drawn from the same distribution (Haussler, 1992). This is significant for building reliable and practical ML models. Usually, this ability is measured by the model's accuracy on the testing set (not involved in model training and validation).

Reference:

Haussler, D.: Decision Theoretic Generalizations of the PAC Model for Neural Net and Other Learning Applications, Inf. Comput., 100, 78–150, https://doi.org/10.1201/9780429492525-4, 1992.

(8) (l. 66) "the stacking strategy"

Please elaborate on what the stacking strategy means.

**Reply:** Thanks for your comment. We briefly elaborated the meaning of stacking strategy in the beginning of Section 3.3.3 in our original manuscript. Per your kind suggestion, we moved it to the corresponding position of Introduction to make the comcept of stacking strategy being more naturally introduced. (Lines 71–73)

(9) (l. 72) "the physical observations"

Again, it is unclear what they are.

**Reply:** Good suggestion! We merged this sentence with the next one to specify what

physical observations they are. (Lines 79–81)

(10)(l. 74) "vertical ion drift (VIC)"

This should be "vertical ion convergence (VIC)".

**Reply:** Thanks. Amended as suggested. (Line 81)

(11)(l. 78) "the meta model"

The concept of meta model should be explained.

**Reply:** Yes. We described the concept and function of meta model in SML method. (Lines 85–86)

(12)(l. 79) "from different aspects"

Please clarify what "different aspects" refer to.

**Reply:** Point has been well taken. We specified the different aspects as under different space weather conditions. (Line 87)

(13)Figure 1

What are the blue lines?

**Reply:** Thanks for your question. The blue lines represent the traces of COSMIC TEC profiles. We added this in the figure caption of Figure 1.

(14)(l. 97) "remove the profiles with negative TEC values and the bottom heights higher than 90 km"

Roughly how many percent of the data are rejected in this quality control process? Are they already removed in Figure 1?

**Reply:** Good question. There are 93.45% of raw TEC profiles passed the quality control process, and the rejected profiles have been removed in Figure 1. Inspired by your comment, we moved the description of quality control process from the second paragraph of Section 2.1 to the first paragraph to make the text more logistic. (Lines 99–100, Line 104)

(15)Figure 2

It appears that the numbers following the station names indicate the years of

measurements. If so, please clarify this explicitly.

**Reply:** Thanks for pointing this issue out. We clarified this by adding the description in Figure 2 caption as "In the bracket behind each station name, the corresponding time period of data is presented in the format of yy-yy". (Line 122)

(16)(l. 278) "ground truth"

What are "Ground Truth" shown in Figure 8? Are they grid averages over many years including different seasons and local times? Please clarify.

**Reply:** Thank you. According to your suggestion, we clarified that the EsI maps (include ground truth and model-predicted EsI) represent the average EsI in the latitude-longitude bin of $2.5°\times5°$ over the period of testing set. (Lines 287–288)

(17)(l. 310) "The clearly reconstruction of these two ..."

I suggest the authors rewrite this sentence; the wording sounds a bit awkward (e.g. "two larger EsI with different reasons" and "SML model can simultaneously consider different physical mechanisms").

**Reply:** Good suggestion! To avoid confusion, we adopted your suggestion and rewrote the sentence as "It indicates that SML model can clearly reconstruct and predict the larger EsI in mid and high latitudes dominated by different physical mechanisms, and can comprehensively consider the impact of multiple influencing factors." (Lines 318–319)

(18)(l. 353) "Figure 12"

State that Figure 12 represents the results for quiet periods.

**Reply:** Thanks. Amended as suggested. (Lines 371, 387)

(19)(l. 367) "Compared to quiet days, the EsI distributions during storm days are more complex"

It is not clear if this is associated with geomagnetic storms or other factors (season, F10.7, VIC, etc.). If the authors want to demonstrate the impact of geomagnetic storms, they could run the SML model with a fixed Dst value of 0 and compare the results to those obtained using variable Dst values.

**Reply:** Thanks for your comment. As you mentioned, actually, the variations and disturbances of Es layers during geomagnetic storm may be attributed to not only geomagnetic activity, but also a combined effect of complex physical and chemical processes related to wind, electric field, gravity waves, and so on, which have been

stated in the start of Section 4.3. Since the focus of this manuscript is on the prediction performance of SML model during geomagnetic storm rather than the impact of geomagnetic storm, we would not like to demonstrate the impact of geomagnetic storm on EsI distribution. Nevertheless, we tried to run SML model with a fixed Dst value of 0, as shown in the figure below. We can see that the predicted EsI by SML with Dst of 0 is considerably larger than that by SML with variable Dst on Dec 07, 2014 (moderate storm, Dst = -43 nT). This result is consistent with that reported by Liu et al. (2022) that EsI usually has a decrease during moderate storms than during quiet time. Even though we did not demonstrate this result in our manuscript, the method you proposed is effective for studying the response of EsI during different geomagnetic storms, and we will concern about it in the future.

[Figure]

**Figure R1:Latitude-longitude and latitudinal distributions of ground truth, EsI predicted by SML with fixed Dst index of 0, and the corresponding error maps during geomagnetic storm time on 07 Dec 2014 and 22 Jun 2015.**

Reference:

Liu, Y., Zhou, C., Xu, T., Deng, Z., Du, Z., Lan, T., Tang, Q., Zhu, Y., Wang, Z., and Zhao, Z.: Geomagnetic and Solar Dependencies of Midlatitude E-Region Irregularity Occurrence Rate: A Climatology Based on Wuhan VHF Radar Observations, J. Geophys. Res. Sp. Phys., 127, e2021JA029597, https://doi.org/10.1029/2021JA029597, 2022.

(20) Figure 14

Does this include all the ionosonde data from different latitudes? The results seem to show that Smax values are sometimes much greater than those anticipated from foEs.

Can the authors comment on whether data from certain ionosondes are responsible for those discrepancies?

**Reply:** Yes, it includes all ionosonde data from different latitudes. Following your suggestion, we further inspected the data and found that the several data pairs with much larger Smax values are from SO166. This discrepancy may be attributed to its higher latitude (67.4 °N), where RO-based detection of Es layers may have accuracy degradation due to small amount of data or horizontal gradient of ionosphere. In our previous work, Hu et al. (2022), there was also a high-latitude ionosonde station MW26P (67.6 °S) with much lower correlation coefficient between RO Smax and foEs, and the similar cases are also presented in Niu et al. (2019) and Niu and Fang (2023).

References:

Hu, T., Luo, J., and Xu, X.: Deriving Ionospheric Sporadic E Intensity From FORMOSAT-3/COSMIC and FY-3C Radio Occultation Measurements, Sp. Weather, 20, e2022SW003214, https://doi.org/10.1029/2022SW003214, 2022.

Niu, J., Weng, L., and Fang, H.: An attempt to inverse the ionospheric sporadic-E layer critical frequency based on the COSMIC radio occultation data, Adv. Sp. Res., 63, 1204–1213, https://doi.org/10.1016/j.asr.2018.10.029, 2019.

Niu, J. and Fang, H.: An Empirical Model of the Sporadic E Layer Intensity Based on COSMIC Radio Occultation Observations, Sp. Weather, 21, e2022SW003280, https://doi.org/10.1029/2022SW003280, 2023.

(21) Figure 15

It should be stated that in contrast to Figure 14, the model-data comparisons presented in Figure 15 do not incorporate the spatiotemporal window of (0.5 °, 0.5 °, 1 h).

**Reply:** Thanks. Amended as suggested. (Lines 406–407)

(22) Figure 15

Why are these comparisons limited to the three mid-latitude stations in the Northern Hemisphere? Since the authors claim that the model can capture Es variability caused by different mechanisms, the comparisons should be extended to include high-latitude stations.

**Reply:** Agreed! According to your suggestion, we made use of two other ionosonde stations, DW41K (low latitude) and SO166 (high latitude) to replace WU430 and MH453, so that Figure 15 can comprehensively show the evaluation results in all of low, mid and high latitudes. After replacement, the correlation coefficients (CCs) of the three stations are 0.613, 0.739 and 0.636, respectively.

(23) Table 1

This is an interesting evaluation. However, the current approach to assessing the importance of input variables (VIC, GW, F10.7, and Dst) limits the insights that can be drawn. I suggest that the authors evaluate the contribution of each input variable individually by systematically removing them one at a time. As it stands, I am not fully convinced that the GW and Dst parameters meaningfully enhance the model's performance.

**Reply:** Good points! Per your kind suggestion, we modified the combinations of input variables for this evaluation by removing them one by one. The sequence of removal is Dst, F10.7, GW Ep and VIC. As you stated, we found that the increases of ME and RMSE after each removal, which represent the contribution of the corresponding parameter, are larger for VIC and F10.7, and smaller for GW Ep and Dst. We listed some possible reasons for explanation, and we also added the summary at the end of this paragraph. (Lines 416–435)

(24) (l. 431) "Tian et al. (2023) conducted ..."

Could this be more specific? If Tian et al. (2023) did not include VIC, what lower atmospheric parameters did they use?

**Reply:** Thanks. We listed the lower atmospheric parameters they included in their model. (Lines 450–453)

(25) (l. 433) "the physical observations"

Please specify exactly what they are.

**Reply:** Amended as suggested. (Lines 454–455)

(26) (l. 456) "with the improvements of 20.1%-40.5%"

State that these numbers represent the improvements in RMSE.

**Reply:** Amended as suggested. (Line 477)

**Reply to Anonymous Reviewer #2**

Paper Title: "Global ionospheric sporadic E intensity prediction from GNSS RO using a novel stacking machine learning method incorporated with physical observations".

Authors: T. Hu, X. Xu, J. Luo, J. Hou, H. Liu

Preprint: egusphere-2025-1549

**Summary comment**

The article is highly well written and discussed and is almost ready for publication. However, I have a few comments.

**Reply:** Thanks for your kindly evaluation on our manuscript, as well as your constructive suggestions and comments. We revised the manuscript according to your recommendations, and the revisions also took into consideration the comments from another reviewer.

The important revision we made following your comments is that we added the classification of Es layers caused by the factors you mentioned to the introduction and supplemented the relevant literatures. In addition, we explained in detail why we use foEs rather than fbEs and use HWM rather than GSWM in this reviewer response letter.

In this reviewer response letter, your remarks are responded point-by-point. In the revision, all the changes made on the original manuscript have been highlighted in red color.

**Detailed Comments**

(1) The introduction to the Es layer lacks clarity. Globally, Es layers are classified not only based on wind shear and gravity waves, but also include layers associated with irregularities and particle precipitation (Resende et al., 2013; 2016; 2018; 2021; 2024; Moro et al., 2023; 2025). It would be beneficial to incorporate this broader classification into the introduction.

**Reply:** Thanks for your constructive suggestion. We added the classification of Es layers caused by the factors you mentioned to the introduction and supplemented the relevant literatures. (Lines 30, 33–36)

(2) It's worth questioning why the authors chose to use foEs instead of fbEs. The fbEs parameter is often considered more appropriate for characterizing Es layers, particularly in cases where the layer becomes dense enough to block reflections from higher altitudes, indicating a thickening of the layer. Including a justification for the

choice of parameter or considering the use of fbEs.

**Reply:** Thanks for your question. We noticed that fbEs is taken by some recent researches to be a better estimate of Es layer intensity. Emmons et al. (2023) indicated that fbEs has stronger correlations with RO-derived Es intensity than foEs. However, there are fewer fbEs measurements provided by globally distributed ionosondes, and not all Es layers are blanketing, which places a limitation on model construction using fbEs. In our manuscript, the ionosonde stations from NESSDC do not provide fbEs measurements, and not all ionosonde stations from UKSSDC provide fbEs measurements. In addition, ionosonde foEs/fbEs is not used for SML model construction in this study, but only for the validation of SML predictions. Therefore, the choice of foEs or fbEs does not have impact on the prediction performance of SML model. Emmons et al. (2023) compared fbEs and foEs observed by global ionosondes from GIRO, and they found that most fbEs values are close in magnitude to foEs values. Based on their results, we can believe that using foEs can verify the good performance of SML. However, the issue you mentioned is still worth thinking. We will concern about it in the future by gathering more fbEs measurements and comparing it with foEs to verify the superiority of fbEs in characterizing Es layers.

Reference:

Emmons, D. J., Wu, D. L., Swarnalingam, N., Ali, A. F., Ellis, J. A., Fitch, K. E., and Obenberger, K. S.: Improved models for estimating sporadic-E intensity from GNSS radio occultation measurements, Front. Astron. Sp. Sci., 10, 1327979, https://doi.org/10.3389/fspas.2023.1327979, 2023.

(3) Why did the authors use HWM? The GSWM model better represents wind shear at heights below 110 km.

**Reply:** There are two factors that preventing us from using GSWM model. First, GSWM produces monthly migrating tidal climatologies (including winds), which cannot meet the requirement of the higher temporal resolution Es intensity prediction in our manuscript. Second, the upper boundary of GSWM is 120 km, while the altitude range of Es layers considered in our manuscript is 90–130 km. In comparison, HWM14 describes the neutral wind fields from the surface to the exobase (~450 km), which covers the altitude range we consider.

In fact, in addition to HWM14 and GSWM, numerical models which consider a variety of physical and chemical processes can provide neutral wind simulations that are closer to actual observations, such as WACCM-X and TIEGCM. In the work of this manuscript, we were unable to use numerical model due to the limitations in code permissions and computing resources. It can be expected to use numerical models such as SD-WACCMX in the future to simulate the behavior of neutral winds for better prediction of Es intensity.

(4) South American Magnetic Anomaly instead of South Atlantic Anomaly.

**Reply:** Thanks. We modified all such expressions and abbreviations.